# Generative Artificial Intelligence as a Tool for Teaching Communication in Nutrition and Dietetics Education—A Novel Education Innovation

**DOI:** 10.3390/nu16070914

**Published:** 2024-03-22

**Authors:** Lisa A. Barker, Joel D. Moore, Helmy A. Cook

**Affiliations:** 1Department of Nutrition, Dietetics and Food, Monash University, Notting Hill, VIC 3168, Australia; 2School of Social Sciences, Monash University, Melbourne, VIC 3800, Australia; joel.moore@monash.edu; 3School of Medicine, Monash University, Clayton, VIC 3800, Australia; helmy.cook@monash.edu

**Keywords:** artificial intelligence, education, communication, simulated patients, dietetics

## Abstract

Although effective communication is fundamental to nutrition and dietetics practice, providing novice practitioners with efficacious training remains a challenge. Traditionally, human simulated patients have been utilised in health professions training, however their use and development can be cost and time prohibitive. Presented here is a platform the authors have created that allows students to interact with virtual simulated patients to practise and develop their communication skills. Leveraging the structured incorporation of large language models, it is designed by pedagogical content experts and comprises individual cases based on curricula and student needs. It is targeted towards the practice of rapport building, asking of difficult questions, paraphrasing and mistake making, all of which are essential to learning. Students appreciate the individualised and immediate feedback based on validated communication tools that encourage self-reflection and improvement. Early trials have shown students are enthusiastic about this platform, however further investigations are required to determine its impact as an experiential communication skills tool. This platform harnesses the power of artificial intelligence to bridge the gap between theory and practice in communication skills training, requiring significantly reduced costs and resources than traditional simulated patient encounters.

## 1. Introduction

Effective communication skills are fundamental for nutrition and dietetics practitioners and are therefore an essential part of the education of future dietitians, yet little is known about the best approaches to practice communication skills for dietitians [1]. Human simulated patients play a role in supporting the development of skills but are costly and resource intensive. There is a need to understand how innovative, cost-effective approaches support communication skill development in order to advance dietetics education. 

Clear communication between patients and healthcare professionals is essential to promote behaviour change, reduce patient anxiety and build knowledge and trust [2]. Communication skill development for nutrition and dietetics practitioners should focus on rapport building, paraphrasing, empathy, clinical reasoning and active listening [3,4,5]. Utilising these communication skills in a way that is patient centred has been shown to increase patient and practitioner satisfaction and therapeutic relationships, as well as increase patient quality of life and improve clinical outcomes [4,6,7,8].

Incorporating high quality communication training into nutrition and dietetics curriculum design is essential to ensure graduates are well prepared for practice. This is supported through international competency standards and curricula, where communication skills are identified as one of the most frequently included concepts for competency and proficiency [1]. There is also greater recognition of the need for graduates to be able to communicate with a diverse population, for educators to be aware of the diversity within their student population, and the implications of both of these on communication skills training [9].

Traditional approaches to communication education and curricula design in nutrition and dietetics usually include a combination of experiential and didactic learning methods [3,10,11]. Didactic methods include lectures, demonstrations and case-based learning. Experiential methods include role-plays and the use of simulated patients. Human simulated patients (HSPs) are considered the gold standard for authentic clinical encounters [12], however are often used sparingly due to cost and complexity of the organisation process [13]. Additionally, HSPs are commonly minimally-trained actors with a high turnover, so maintaining consistency as well as diversity in patient portrayals can be difficult [14]. Students, such as Janice described in Figure 1, complain about the limited opportunities to work with HSPs, the limited feedback they receive and the inability for repeated attempts [14].

Recently, there has been much interest in the role virtual simulated patients (VSPs) may play in health professions education, as well as emerging evidence that they can improve communication skills, including clinical reasoning [15]. Until now, there has been considerable effort and expense relating to the development of VSPs [16]. However, the release of large language models to the public, for example, ChatGPT, shows great promise for mobilising artificial intelligence in VSP applications in education [17]. This paper describes an example of a novel educational innovation platform to support the development of communication skills in dietetics education. The platform is called the Authentic Teaching and Learning Application Simulation (ATLAS), which was developed by the authors to provide students with authentic learning experiences that allow multiple attempts and instant personalised feedback.

## 2. Method

ATLAS is a voice to chat based platform that harnesses state-of-the-art large language models, artificial intelligence and educator designed patient personas to simulate real-world HSPs. The interface operates through a series of interconnected processes, beginning with the student’s spoken input. This input is captured and translated into text using speech-to-text algorithms. The text is then fed, together with detailed persona data and the conversational history, into a large language model. Next, the model generates a contextually appropriate response that moves the conversation logically forward in a way that aligns with the persona’s characteristics and the educational objectives of the simulation. Finally, the response is returned to the student as synthesised speech, together with a textual description of the patient’s body language. This unique platform was created to replicate learning from HSP interactions multiple times during course progression with varying cases and personas. This interface is shown pictorially in Figure 2. 

The ATLAS platform allows pedagogical content experts to design individual cases based on curricula and student needs. For the learner, the experience is an immersive, authentic, professional interaction that tests their ability to apply knowledge and skills they have learned in a dynamic, unscripted conversation. Interaction with the VSP via voice recognition represents a more realistic student–patient interaction, requiring students to articulate their message professionally, with empathy and clarity. 

In order to mimic real-life interactions, development of patient personas with information similar to what would be required for HSPs was required: age, occupation, current health status and past medical history, as well as information they should share willingly, and information they should only share if asked directly. VSPs were further developed by outlining a personal communication style (for example chatty, direct, anxious, disengaged), replicating the diversity observed in the real world. Additionally, VSPs were trained to respond similarly to an HSP; if rapport was not developed or questions that might disengage a real patient are asked (for example, if the student touches on a sensitive subject without showing empathy), the VSP would ‘shut down’ and no longer provide detailed answers. Body language was indicated in parentheses, for example, “(looks around uncomfortably)”, to help students understand the importance of non-verbal communication.

## 3. Results

During conversations, students found they were required to build rapport, just as they would with a real-world patient. Without this, answers from VSPs remain short and without detail. As topics arise that are often considered uncomfortable to discuss (for example alcohol use), students can take the opportunity to react to patient responses or body language, phrase and paraphrase questions and responses, and make mistakes which are essential to their learning. Students can also take their time to perfect their diet history taking skills, collecting detailed food quantities to allow thorough dietary analysis. As this question-and-answer process can come across as monotonous with novice practitioners, and can take upwards of 30 min, students appreciated being able to utilise this platform outside of the classroom, bringing additional benefits to help them refine this key skill. An excerpt of an interaction between a student dietitian and an ATLAS persona is provided in Figure 3.

ATLAS was able to provide feedback based on a validated nutrition and dietetics communication assessment tool [18] analysing inputs against the expectations for communication (e.g., open-ended questions), to provide detailed written feedback for student learning (Figure 4). This occurs at the end of every conversation, which encourages the student to self-reflect. Students appreciated this and took the opportunity to undertake repeat attempts and use the platform outside of the classroom, providing unlimited opportunities to reattempt and improve performance.

## 4. Discussion

The ATLAS platform is being utilised throughout the Master of Nutrition and Dietetics program at Monash University, with students having opportunities to converse with ATLAS VSPs at various timepoints during their degree. As with all new educational tools, it is important that it is embedded in the existing communication skills curriculum [19]. Students have accessed the platform after being prepared with theory on communication, but before (and after) HSP and placement opportunities. The platform allows students to practice difficult conversations, such as collecting information on usual bowel habits, or collecting detailed information such as a weight and diet histories. While it Is known that experiential communication skills training is particularly important for increasing student confidence in understanding the patient perspective [3], this platform is currently being evaluated to understand its impact in this regard. The platform has been developed so the VSP has memory of earlier conversations with individual students, so students can pick up where they left off, as well as a reset button so that students can clear the memory of previous discussions. This may allow deeper rapport development with the patient, as well as the ability to repeat conversations with the same VSP to practice the wording of certain phrases and improve written feedback. Opportunities also exist for platform utilisation during assessments and examinations, which are currently being explored by the teaching team. Cross-faculty collaboration has occurred within our university, with the platform being applied to contexts in medicine, education, science and art faculties to practice professional communication.

Although this technology is rapidly developing, limitations to VSPs do exist. A broader diversity of patient portrayals has not yet been researched to assess cultural competence and safety. Learning experiences should not reinforce biases, therefore AI identities should be three dimensional and avoid stereotyping [20]. Having the personas designed and tested by people with a shared lived experience may ensure this, and reduce the bias inherent in commercial large language models. This is in line with patient-centred care principles, which calls for patient involvement in education design [21]. Further research in this area is required. There is also a need to ensure that simulations are pedagogically sound and are accompanied by measures of quality and impact on learning to build evidence that harnessing generative artificial intelligence as a communication tool in health professions education is effective. For example, the ATLAS VSPs should be compared with other active learning methods [22], with such a study currently being undertaken by these authors. Finally, there are important ethical considerations in the development of AI technology, such as data protection for students [23].

## 5. Conclusions

The use of AI to support nutrition and dietetic students’ healthcare communication is a new area in communication skills curricula. The uniquely developed ATLAS platform has many key strengths in its design, and its incorporation into the curriculum in a scaffolded manner allows for enhanced communication skill practice and greater educator and learner flexibility. Its adaptability means interdisciplinary application beyond nutrition and dietetics is straightforward. This platform not only bridges the gap between theory and practice in communication skills training, but also the gap between generative AI, academics and experienced practitioners.

## Figures and Tables

**Figure 1 nutrients-16-00914-f001:**
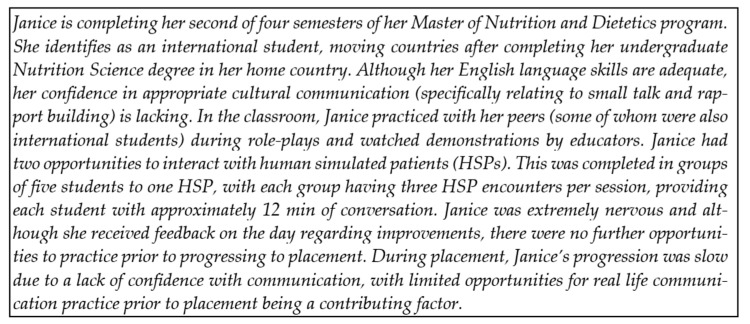
Representative student, Janice (not her real name), needing additional support with communication training.

**Figure 2 nutrients-16-00914-f002:**
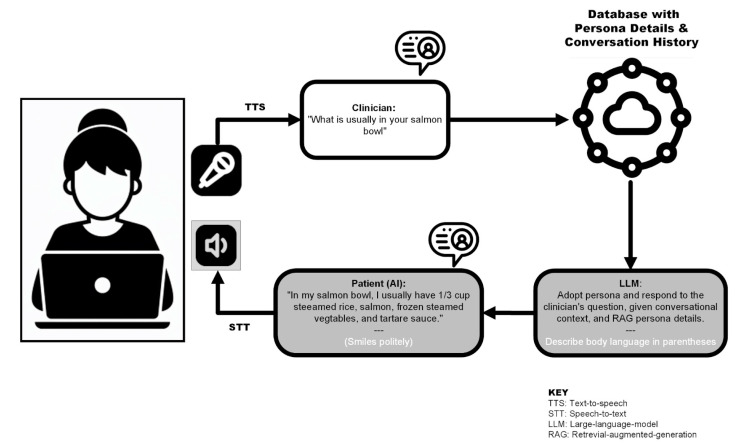
The operational interface of the ATLAS platform.

**Figure 3 nutrients-16-00914-f003:**
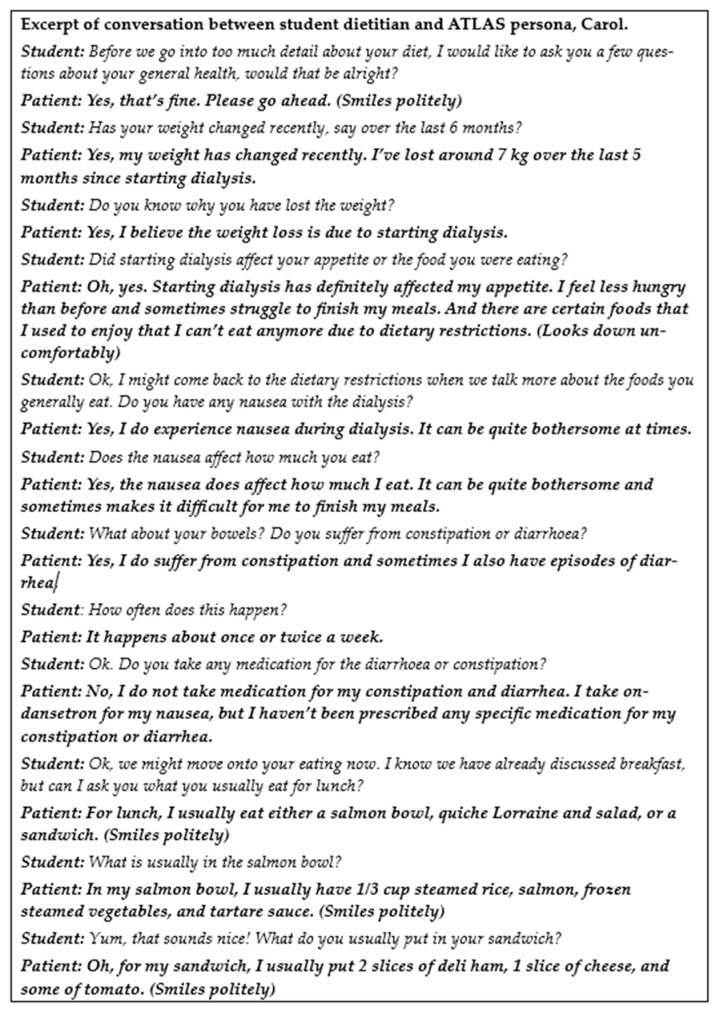
Excerpt of conversation between student dietitian and ATLAS persona, Carol.

**Figure 4 nutrients-16-00914-f004:**
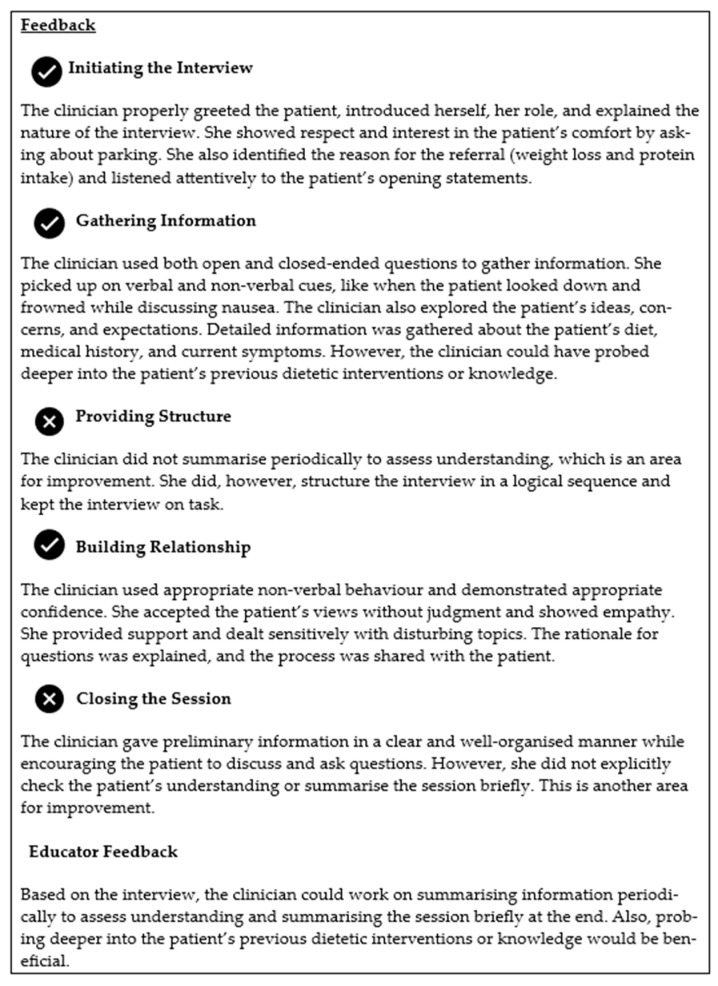
An example of feedback generated for students within the ATLAS platform.

## Data Availability

The raw data supporting the conclusions of this article will be made available by the authors on request.

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
