# Peer review of "Generative Artificial Intelligence as a Tool for Teaching Communication in Nutrition and Dietetics Education—A Novel Education Innovation"

_nutrients, 2024, doi:10.3390/nu16070914_

Round 1
Reviewer 1 Report
Comments and Suggestions for Authors
Barker and the co-authors have submitted a brief communication that discusses the utilization of Generative Artificial Intelligence as a tool for teaching communication in the field of nutrition and dietetics education. Effective communication skills are crucial for the practice of nutrition and dietetics, and therefore, dietetic educators require innovative, cost-effective approaches to support communication skill development. The authors describe the development and implementation of a novel educational innovation platform called Authentic Teaching and Learning Application Simulations (ATLAS) to support the development of communication skills in dietetics education.
While this is not an original research article, the intention appears to be twofold.
First, through describing the superior operations of platform, including immediate feedback, it is intended to encourage other educators to consider ATLAS platform as a viable option for their programs. However, the paper does not provide information that would allow other programs to adapt and build this into their programs. Most specifically, there is no reference for ATLAS given in the paper. Please add a reference to the tool used in paper, or describe how other programs could replicate this work.
Second, this lays the groundwork for future studies invesitagating learning outcomes among dietetic students, which has yet to be documented. There was clear mention on the need to build evidence as an effective education tool in lines 222-226. Yet, the statement on line 232 "allows for deeper learning" is not yet supported. Please revise this statement.
Author Response
The authors thank the reviewer for their suggestions in how our manuscript could be strengthened. Please see below for how we have incorporated these changes.
1. Please add a reference to the tool used in paper, or describe how other programs could replicate this work.
The ATLAS platform was created by the authors of this paper and is currently in final evaluation stages. A subsequent paper will describe in more details the full evaluation process and results once this is complete. We have added a sentence in the methods to make this clearer (line 94) and also added a flow diagram which more clearly describes the way the platform was created (Figure 2).
2. There was clear mention on the need to build evidence as an effective education tool in lines 222-226. Yet, the statement on line 232 "allows for deeper learning" is not yet supported. Please revise this statement.
This has been changed to “allows enhanced communication skill practice”.
Reviewer 2 Report
Comments and Suggestions for Authors
Although clear communication is fundamental to nutrition and dietetics practice, little is known about approaches to efficacious training for novice practitioners. Traditionally, human simulated patients have been utilised in health professionals training, including in nutrition and dietetics education. By carefully modelling patient-clinician interactions and leveraging structured incorporation of large language models, AUTHORS have created a platform where students can interact with virtual simulated patients to practise and develop communication skills. This voice to chat based platform comprising individual cases based on curricula and student needs and designed by pedagogical content experts allows for targeted practice of rapport building, asking of difficult questions, paraphrasing and mistake making, all of which are essential to learning.
THEIR platform also provides students with individualised feedback based on validated communication tools to encourage self-reflection and improvement.
AUTHORS conclude that THEIR platform harnesses the power of artificial intelligence to bridge the gap between theory and practice in communication skills training, requiring significantly reduced costs and resources than traditional simulated patient encounters.
Interesting study facing an emerging topic.
I have only some suggestions for the authors.
1. Abstract could be improved better summarizing the sections
2. Section 2 contains introductive discourses with references please modify
3. I suggest to insert a flow chart to describe the design in the methodological description and describe the design following it.
4. Results, discussion and conclusions must be separated and expanded according to the standards
Author Response
The authors sincerly thank the reviewer for the suggested improvements to the manuscript. These have been incorporated per the points below.
- Abstract could be improved better summarizing the sections
The abstract has been re-written with a focus on summarizing the sections of the manuscript more clearly.
- Section 2 contains introductive discourses with references please modify
The introductory sentence beginning at line 102 has been deleted and incorporated into the introduction. There is one additional reference in the methodology section, however it refers to the validated feedback tool being described in the methods section.
- I suggest to insert a flow chart to describe the design in the methodological description and describe the design following it. Joel,
Thank you for this suggestion, This has been added to the methods section as Figure 2, as well as some additional text in the methods to describe the platform development and operation.
- Results, discussion and conclusions must be separated and expanded according to the standards
The headings of the manuscript have been amended to include results, discussion and conclusion, and have been expanded per the reviewers recommendations.
Reviewer 3 Report
Comments and Suggestions for Authors
Well written and very timely. Appreciate the examples included in Figures 1, 2, and three. This type of learning program is very much needed for dietetic interns and nutrition graduate students. I'm sure there will be interest in this type of tool on a broad scale for dietetics education and education of other health professional students.
Author Response
The authors thank the reviewer for the positive comments made in relation to the platform we have developed and its application in dietetics communication training.